# Temporal Distance-aware Transition Augmentation for Offline Model-based Reinforcement Learning

**Dongsu Lee** [1 2]  **Minhae Kwon** [2 3]

## Abstract

The goal of offline reinforcement learning (RL) is to extract a high-performance policy from the fixed datasets, minimizing performance degradation due to *out-of-distribution* (OOD) samples. Offline model-based RL (MBRL) is a promising approach that ameliorates OOD issues by enriching state-action transitions with augmentations synthesized via a learned dynamics model. Unfortunately, seminal offline MBRL methods often struggle in sparse-reward, long-horizon tasks. In this work, we introduce a novel MBRL framework, dubbed **Temp**oral **D**istance-**A**ware **T**ransition **A**ugmentation (**TempDATA**), that generates augmented transitions in a temporally structured latent space rather than in raw state space. To model long-horizon behavior, TempDATA learns a latent abstraction that captures a *temporal distance* from both *trajectory and transition levels* of state space. Our experiments confirm that TempDATA outperforms previous offline MBRL methods and achieves matching or surpassing the performance of diffusion-based trajectory augmentation and goal-conditioned RL on the D4RL AntMaze, FrankaKitchen, CALVIN, and pixel-based FrankaKitchen.

https://dongsuleetech.github.io/
projects/tempdata/

## 1. Introduction

RL has long been recognized as a powerful paradigm for sequential decision-making. However, it is hindered in real-world applications by its reliance on trial-and-error

This work was partly done at [1]Carnegie Mellon University, Pittsburgh, USA [2]Department of Intelligent Semiconductors, Soongsil University, Seoul, South Korea [3]School of Electronic Engineering, Soongsil University, Seoul, South Korea. Correspondence to: Minhae Kwon <minhae@ssu.ac.kr>.

*Proceedings of the $42^{nd}$ International Conference on Machine Learning*, Vancouver, Canada. PMLR 267, 2025. Copyright 2025 by the author(s).

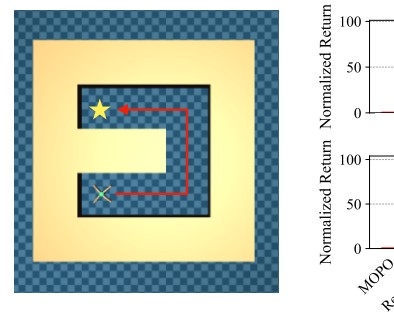 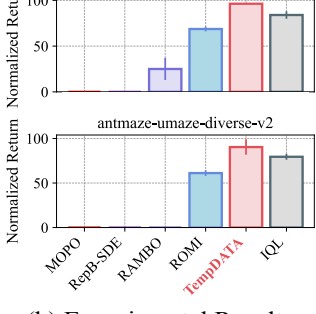

(a) AntMaze Umaze     (b) Experimental Results

*Figure 1.* **Performance comparison overview**. (a) Umaze environment, the most naive level among AntMaze. The 8-DoF ant robot navigates the maze to reach the goal state, marked as a yellow star. (b) Comparison between the proposed solution and previous MBRL on two D4RL benchmark datasets. TempDATA (proposed) achieves the best performance in two benchmarks.

interactions. The offline paradigm allows RL to control its limitations by leveraging the offline dataset without real-time interactions, achieving notable successes in several domains (Snell et al., 2023; Tang et al., 2022; Lee et al., 2024a; Lee & Kwon, 2023; 2025; Eo et al., 2023).

Previous off-policy RL often suffers from evaluating OOD actions taken by the learned policy when estimating a state-action value. Although online RL does not need to consider OOD, offline setups cannot consider additional data through online interactions. Therefore, offline RL aims to extract the best possible policy from the fixed offline dataset by considering how to handle OOD. One promising workaround is offline model-free RL (MFRL), which imposes conservative regularization on policies or value functions toward the distribution of state–action pairs in the offline data (Kumar et al., 2020; Kostrikov et al., 2022).

An alternative option is offline MBRL methods that learn a dynamic model. They alleviate the OOD issue by synthesizing new transitions via planning with the learned dynamics model and a behavior policy (Yu et al., 2020; 2021; Rigter et al., 2022; Sun et al., 2023). These methods have covered OOD samples efficiently, achieving better performance than offline MFRL baselines on dense-reward, short-horizon

robotic tasks (*i.e.*, D4RL Gym Locomotion). However, it has been observed that most offline MBRL methods can be problematic in sparse-reward and long-horizon environments, related to goal-achieving tasks. Our empirical evaluation confirms near-zero success rates for MBRL baselines on the canonical AntMaze benchmark, which is a representative goal-achieving environment. Even on the simpler *umaze* variant, most MBRL methods fail as shown in *Figure* 1.

Unlike the MFRL (IQL by Kostrikov et al. in *Figure* 1), offline MBRL stems from three core issues: over-generalization in out-of-support areas (*e.g.*, obstacles, walls), a biased model-based dataset, or insufficient learning signals about long-horizon behaviors (Wang et al., 2021; Wu et al., 2024). Recent offline MBRL learns an adversarial or a reverse dynamic model to avoid over-generalization, *e.g.*, RAMBO by Rigter et al. and ROMI by Wang et al. in *Figure* 1. However, they still struggle to synthesize transitions that incorporate temporal information, which is how distant apart from a goal state, to learn long-horizon behavior.

This work aims to tackle sparse-reward and long-horizon RL tasks via offline MBRL. To do this, we introduce a novel MBRL framework, dubbed TempDATA. TempDATA consists of three components: an autoencoder, a latent dynamics model, and an offline policy. The autoencoder has two objectives: 1) embed the temporal distance in terms of both trajectory and transition levels as a representation, and 2) reconstruct original states from latent representations. The latent dynamic model generates additional transitions in a latent space, mitigating overgeneralization. Moreover, offline policy is trained by skill RL or goal-conditioned RL (GCRL) techniques. This design enables more efficient augmentation than operating in high-dimensional state spaces.

In our experiments, we assess TempDATA, a suite of goal-oriented benchmarks, covering state-based domains, *e.g.*, AntMaze (Brockman et al., 2016), Kitchen (Gupta et al., 2019), CALVIN (Mees et al., 2022), and a pixel-based Kitchen variant. We confirm that our solution outperforms previous baselines. Furthermore, the proposed solution achieves better or comparable performance compared to prior goal-conditioned MFRL methods. To our knowledge, TempDATA is the first offline MBRL approach to enable efficient transition augmentation for sparse-reward and long-horizon challenges.

## 2. Related Works

**Offline Reinforcement Learning.** Offline RL has two branches for solving MDP: model-free and model-based solutions. Offline MFRL algorithms regularize the distribution of the policy and $Q$ function to be close to the distribution of the dataset. For example, behavioral-regularized approaches explicitly constrain policy distribution using KL divergence (Wu et al., 2019; Zhou et al., 2021), MMD kernel (Kumar et al., 2019; Zhang et al., 2021), Wasserstein distance (Ma et al., 2021), and BC loss (Fujimoto & Gu, 2021; Tarasov et al., 2023). Other methods implicitly regularize via weighted regression (Nair et al., 2020; Mitchell et al., 2021), conservative $Q$-learning (CQL) (Kumar et al., 2020), uncertainty quantification to the $Q$ estimations (An et al., 2021), and next-action query avoidance (Kostrikov et al., 2022; Xu et al., 2023). Different from these methods, the proposed solution augments the dataset by leveraging the learned dynamic model with the offline dataset.

Next, Offline MBRL algorithms involve learning a model of the environment via supervised learning (SL) with maximum log-likelihood estimation of the MDP, and then generating transition data using the learned model (Yu et al., 2020; Kidambi et al., 2020; Argenson & Arnold, 2021; Yu et al., 2021; Lee et al., 2021; Rigter et al., 2022; Zhang et al., 2023; Rafailov et al., 2023). This approach uses additional data to optimize a policy and alleviate the conservativeness of offline RL, thereby having the potential for better generalization. Although these prior works have shown promising results, they could not solve the goal-reaching tasks and use pixel-based states. This work aims to construct a dynamic model in a representation space that preserves the temporal distance within a state space of any MDP.

**Goal-conditioned Reinforcement Learning.** Early GCRL formulations are primarily tackled using SL approaches (Kaelbling, 1993; Schaul et al., 2015) with pre-defined goals and reward functions to guide the agent toward them. They aim to develop efficient exploration methods (Sukhbaatar et al., 2018) and planning (Nasiriany et al., 2019) for long-horizon tasks. These supervised methods struggle to generalize because they depend on goal-specific supervision. A recent line of GCRL is towards an unsupervised setting, where the goal is to train agents capable of reaching any goal state in the environment without goal-reaching supervision. This unsupervised method aims to learn value functions (Eysenbach et al., 2021; Ghosh et al., 2023) or representations (Wang et al., 2020; Zheng et al., 2024), which can discover goals and corresponding goal-conditioned policies. Several works extract these goal-conditioned networks using skill discovery (Ajay et al., 2021), hierarchical planning (Pertsch et al., 2020; Park et al., 2023a), contrastive learning (Eysenbach et al., 2022), and distance mapping (Myers et al., 2024; Wang et al., 2023).

This work builds a state abstraction that embeds temporal distance with any goal state or next state. Afterward, we optimize a policy by leveraging the representation network and dynamic model in a latent space.

**Learning State Space Abstractions.** Learning representations in RL, referred to as abstraction, aims to map the Markov state space into a latent space, which helps to un-

derstand the state space by capturing the essential features within the MDP. More precisely, it can compress higher-dimensional space (*e.g.*, pixel-based state) as a smaller one (Yarats et al., 2021; Park et al., 2023a; Fujimoto et al., 2023) and extract temporal difference information between two states within any MDP (Pong et al., 2018; Wang et al., 2020). Objectives span successor features (Dayan, 1993; Barreto et al., 2017; Mazoure et al., 2023), contrastive losses (Laskin et al., 2020; Eysenbach et al., 2022; Zheng et al., 2023), metric learning (Liu et al., 2023), dynamics modeling (Seo et al., 2022), bisimulation (Estruch et al., 2022), and distance learning (Hejna et al., 2023).

Particularly, distance learning proves effective for GCRL. Understanding temporal differences provides an intuitive measurement of how transitions between states are difficult. Temporal distance becomes a better mathematical tool than Euclidean distance for tackling real-world problems involving various objects, structural obstacles, and image-based information. Existing studies map distance into Lipschitz (Lecarpentier et al., 2021), metric (Reichlin et al., 2024), quasimetric (Wang et al., 2023), Hilbert (Park et al., 2024), and Riemannian spaces (Tennenholtz & Mannor, 2022). Including triangular inequality in these spaces enhances their utility in efficiently reaching goal states.

This work aims to learn a geometric autoencoder that extracts temporal distance from a raw state and recovers it from a latent state. We design its objectives to reconstruct the raw state from both the micro- and macroscopic levels.

## 3. Preliminaries

**Markov Decision Process (MDP).** An RL problem can be formulated using an MDP, which is defined as a tuple $\mathcal{M} = \langle \mathcal{S}, \mathcal{A}, P, \rho_0, r, \gamma \rangle$. This includes a state $s \in \mathcal{S}$, an action $a_t \in \mathcal{A}$, a state transition probabilities $P : \mathcal{S} \times \mathcal{A} \to \Delta(\mathcal{S})$, an initial state probability $\rho_0 : \Delta(\mathcal{S})$, a reward function $r : \mathcal{S} \to \mathbb{R}$, and a temporal discounted factor $\gamma$.[1] This work casts the MDP as a discrete-time, infinite-horizon, and deterministic environment, where state transition probability maps a state-action pair to the next state (Dekel & Hazan, 2013; Ma et al., 2023).

**Offline RL.** An offline paradigm aims to extract a value function and policy by leveraging a fixed dataset $\mathcal{D}$ collected by a mixture of several behavioral policies. Each behavioral policy $\pi_\beta$ can be optimal, suboptimal, and random for a specific task. An offline dataset $\mathcal{D}$ includes a set of trajectories $\tau = (s_0, a_0, s_1, \cdots, s_H)$, where $H$ is a time horizon of an episode. Each trajectory sample is sampled from trajectory distribution $p^{\pi_\beta}(\tau) = \rho_0(s_0) \prod_{t=0}^{H-1} \pi(a_t|s_t) P(s_{t+1}|s_t, a_t)$. The main objective of an RL is to learn a policy $\pi : \mathcal{S} \to \Delta(\mathcal{A})$ that maximizes

[1]$\Delta(\cdot)$ represents the probability simplex.

the cumulative discounted reward $\mathbb{E}[\sum_{t=0}^{H} \gamma^t r(s_t, a_t)]$. We can extract such a policy using a value-based approach (*i.e.*, $Q$-learning), which approximates a state-action value function $Q(s, a)$ (or state-action value function $V(s)$) with a Bellman optimality operator $\mathcal{B}$ as follows.

$$\mathcal{B}Q(s,a) = \mathbb{E}_{s' \sim P(s'|s,a)}[r + \gamma \arg \max_{a' \sim \pi(s')} Q(s', a')]$$

In practice, value function $Q(s, a)$ (or $V(s)$) is parameterized using a neural network. Such a parameter $\theta$ is optimized by minimizing the temporal-difference error as follows:

$$\mathcal{L}(\theta) = \mathbb{E}_{(s,a,r,s') \sim \mathcal{D}}\big[Q_\phi - \mathcal{B}Q_{\bar{\phi}}\big],$$

where $\bar{\phi}$ means a target network parameter to stabilize the learning process (Mnih et al., 2015). The target network parameter is updated by the Polyak averaging method (Polyak & Juditsky, 1992).

**Offline MBRL.** Offline RL extracts a value function and policy by leveraging a fixed dataset. The model-based solution includes approximating a state transition function to make the most of the fixed dataset. More precisely, this approach builds the learned MDP $\widehat{\mathcal{M}} = \langle \mathcal{S}, \mathcal{A}, \widehat{P}, \rho_0, r, \gamma \rangle$, which $\widehat{P}$ is a learned state transition function that trained using maximum log-likelihood estimation as follows (Yu et al., 2020; Kidambi et al., 2020).

$$\mathcal{L}(\widehat{P}) = \mathbb{E}_{(s,a,s') \sim \mathcal{D}}\big[ - \log \widehat{P}(s'|s,a)\big]$$

Once a model has been learned, MBRL rollouts a transition by leveraging $\widehat{P}(s'|s, a)$ with any state $s \in \mathcal{D}$. Augmented transitions are stored in a separate replay buffer $\widehat{\mathcal{D}}$. Finally, offline MBRL extracts a policy and value function using data sampled from $\mathcal{D} \cup \widehat{\mathcal{D}}$.

**Problem Setting.** This work focuses on goal-reaching tasks. We consider an offline setting with a fixed unlabeled (reward-free) trajectory dataset. We handle such problems using goal state information $s_{\text{goal}} \in \mathcal{G}$, where goal space $\mathcal{G}$ is the same as state space $\mathcal{S}$. The main objective is to reach the goal state $s_{\text{goal}}$, thus reward function can be defined as goal-conditioned reward function $r_g(s) = \mathbb{1}(s = s_{\text{goal}})$ (Nasiriany et al., 2019; Park et al., 2023a).

We aim to augment a state transition dataset useful for achieving a given goal state $g$. Existing MBRL algorithms sometimes synthesize state transitions on inaccessible regions beyond obstacles or cul-de-sacs. To effectively address this, we will show constructing a temporal distance-aware representation $z \in \mathcal{Z}$ of the state $s \in \mathcal{S}$ and then augmenting transitions within the representation space $\mathcal{Z}$.

## 4. TempDATA: Temporal Distance-aware Transition Augmentation

This section introduces TempDATA, our offline model-based scheme that augments new transitions, which help

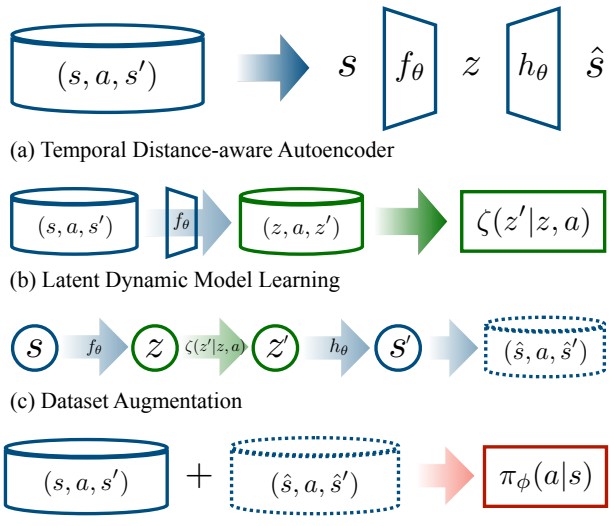

(a) Temporal Distance-aware Autoencoder

(b) Latent Dynamic Model Learning

(c) Dataset Augmentation

(d) Policy Extraction with Offline RL

*Figure 2.* **Illustration for the proposed MBRL framework.** (a) Train autoencoder using offline dataset. (b) Train a dynamic model using a trained encoder and offline dataset. (c) Generate transition dataset using autoencoder and offline dataset. This happens in a representation space. (d) Extract a policy from offline and generated datasets using offline RL algorithm. Processes (c) and (d) are performed iteratively together.

search the pathway to reaching goals. We begin by learning a geometric autoencoder that captures the temporal distance between states of an MDP. Next, we train a transition model in this latent space, allowing rollouts that respect the learned geometry. Finally, we augment the offline dataset with synthetic transitions and train an offline policy engineered for downstream goal-reaching tasks. A graphical overview of TempDATA appears in *Figure* 2.

### 4.1. Temporal Distance-aware Representation

We would like to learn a temporal distance-aware representation encoder $f : \mathcal{S} \to \mathcal{Z}$ from the offline dataset $\mathcal{D}$ which we can later use as a state abstraction $z$ for building transition models and policies to solve downstream goal tasks. Such state abstraction could provide temporal distance to reach the goal beyond spatial distance $||s - s_{\text{goal}}||$. Intuitively, the closeness of the Euclidean space does not simply translate into the temporal distance (*Figure* 3). Next, we need a decoder $h : \mathcal{Z} \to \mathcal{S}$ to recover the augmented transition from the representation space to the state space.

To achieve it, we train the autoencoder by leveraging reconstruction loss (Bank et al., 2023).

$$\mathcal{L}_{rec} = \arg\min_\theta \mathbb{E}_{s\sim\mathcal{D}} \Big[ ||s - h \circ f(s; \theta)|| \Big]$$

Subsequently, we consider geometrical constraint $\mathcal{R}(\theta)$, motivated by (Nazari et al., 2023), where $\theta$ is a parameter of

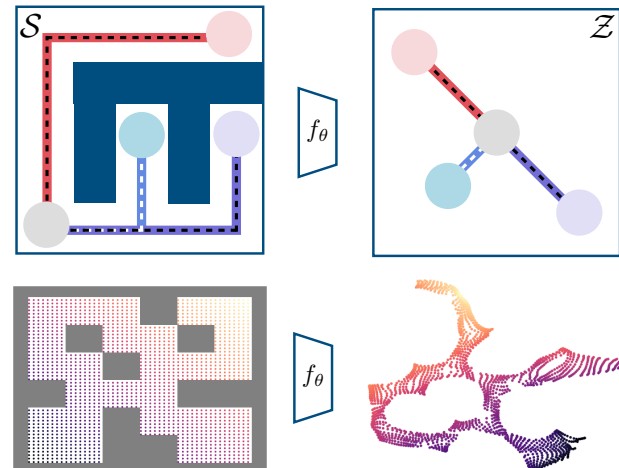

*Figure 3.* **Intuition of state abstraction.** (*Top*) Temporal-aware autoencoder, our main idea, maps into state space into a representation that preserves temporal distance information. (*Down*) An empirical result shows that our encoder can map raw state space as latent state space in antmaze-medium environments. Here, we consider an ant position state of raw state space as fixed and use t-SNE to plot latent state space visually.

the autoencoder. Through such a constraint, the desiderata we want to achieve are as follows.

1. (*Trajectory-level*) Macroscopically, a representation space embeds the temporal distance information of the shortest path between a state and a goal.

2. (*Transition-level*) Microscopically, a temporal distance-aware representation between a state $s$ and a next state $s'$ is less than a pre-defiend distance $d_0$.

The geometrical constraint $\mathcal{R}(\theta)$ for an autoencoder $\theta$ in goal-conditioned RL tasks starts from following **Proposition 4.1** (Wang & Isola, 2022; Wang et al., 2023).

**Proposition 4.1** (Value-metric Equivalence). *Generally, an optimal goal-conditioned value function for a state $s$ and goal $s_{\text{goal}}$ is same with an optimal temporal distance as follows:*

$$V^*(s, s_{\text{goal}}) = -d^*\big(f(s; \theta), f(s_{\text{goal}}; \theta)\big), \quad (1)$$

*where $f(s; \theta)$ is a representation encoder that captures the temporal distance of an MDP on state space $\mathcal{S}$.*

The goal-conditioned value function $V(s, s_{\text{goal}})$ quantifies how quickly a given policy can reach the goal $s_{\text{goal}}$. Therefore, the optimal goal-conditioned value function equals the $Q$-function with an optimal policy.

$$\begin{aligned} V^*(s, s_{\text{goal}}) &\triangleq \max_\pi V^\pi(s, s_{\text{goal}}) \\ &= \max_{a\in\mathcal{A}} Q^*(s, a, s_{\text{goal}}) \end{aligned} \quad (2)$$

Consequently, we regularize the autoencoder such that $z$ preserves a temporal distance $d(f, s, s_{\text{goal}}, \theta)$ of state space. This is achieved by leveraging the temporal differences with the Bellman optimality operator, similar to learning a value function. Additionally, we adopt $r_g(s) - 1$ (*i.e.*, $\mathbb{1}(s \neq s_{\text{goal}})$), instead vanilla goal-conditioned reward function (Kostrikov et al., 2022; Tarasov et al., 2024; Shin et al., 2023; Lee et al., 2024b).[2] Comprehensively, the Bellman optimality target for building temporal distance-aware representation is defined as $\mathcal{B}d = \mathbb{E}\big[(r_g(s) - 1) + \max_\theta \gamma d(f(s'; \theta), f(s_{\text{goal}}; \theta))\big]$. In practice, we use expectile regression (Newey & Powell, 1987) to implement Bellman optimality operator as follows (Kostrikov et al., 2022):

$$\mathcal{L}_{traj} = \mathbb{E}_{\substack{(s,s') \sim \mathcal{D} \\ s_{\text{goal}} \sim p_{\text{goal}}}} \left[ L_2^\tau \Big( \mathcal{B}d - d\big(f(s; \theta), f(s_{\text{goal}}; \theta)\big) \Big) \right],$$

where $L_2^\tau(x) = |\tau - \mathbb{1}(x < 0)|x^2$. Thanks to this formulation, we can extract a representation capturing the shortest path from $s$ to $s_{\text{goal}}$ instead estimate an averaging distance distribution from the probabilistic transitions.

**Theorem 4.2.** *If $\tau = 1$, $\tau$-th representation function-based temporal distance $d\big(f, s, s_{\text{goal}}, \theta\big)$, trained by $\mathcal{L}_{traj}$, is equal to a value function with optimal policy for any $s$ as follows:*

$$\lim_{\tau=1} d_\tau\big(f(s; \theta), f(s_{\text{goal}}; \theta)\big) = -\max_\pi V^\pi(s, s_{\text{goal}}) \quad (3)$$
$$= \text{shortest path from } s \text{ to } s_{\text{goal}}$$

*Proof.* See Appendix B. □

Next, *transition-level* regularizer is related to temporal coherence of consecutive state $(s, s')$. It constrains a representation space to maintain temporal consistency between single-step transitions as $d(s, s') \leq d_0$. The loss function is as follows.

$$\mathcal{L}_{tran} = \mathbb{E}_{(s,s') \sim \mathcal{D}} \left[ L_2^{\tau=1} \Big( d\big(f(s; \theta), f(s'; \theta) - d_0\big) \Big) \right]$$

In this equation, we set $d_0$ as $|r_g(s) - 1|$, akin to a moving cost. This approach is similar to (Allen et al., 2021). They have observed that it encourages the smoothness of representation space and avoids overestimation about a distance of consecutive states.

**Full Objective.** We use the stochastic gradient descent (Ruder, 2016) to update the parameter of autoencoder $\theta$ with the following loss function.

$$\mathcal{L}(\theta) = \mathcal{L}_{rec} + \eta_1 \mathcal{L}_{traj} + \eta_2 \mathcal{L}_{tran} \quad (4)$$

In (4), $\eta_1$ and $\eta_2$ are balancing weights for regularizers. This work considers a deterministic autoencoder, but can be replaced with a variational one (Kingma & Welling, 2013).[3]

---

[2] General methods use transitions by subtracting 1 from $r_g(s)$.

[3] The encoder and decoder can be decoupled for training.

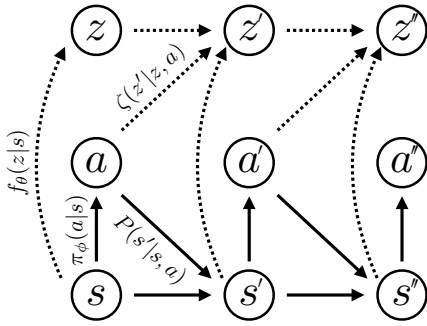

*Figure 4.* **Transitions in a representation space $\mathcal{Z}$.** Dashed lines are related to representation space.

### 4.2. Latent Dynamic Transition Model

This work aims to construct a forward dynamic model in a representation space, dubbed a latent dynamic model, not a state space. Unlike a dynamic model in a state space, the latent dynamic model can handle high-dimensional domains (*e.g.*, pixel-based environments), enhancing generalization and prediction performance.

More precisely, we suppose that a latent dynamic model follows a Gaussian distribution, where both the mean and variance are parameterized by a feed-forward neural network. This model predicts the one-step transition of representation $z \to z'$ by marginalizing out the state space $\mathcal{S}$.

$$\zeta(z'|z, a) = \iint ds ds' p(z'|s') p(z|s) P(s'|s, a)$$

In this equation, we can replace a latent probability $p(z|s)$ as a learned encoder $f(s; \theta)$ in Section 4.1. We parameterize the latent dynamic model $\zeta$ and optimize it via a negative log-likelihood estimate minimization.

$$\mathcal{L}(\zeta) = \mathbb{E}_{\substack{(s,a,s') \sim \mathcal{D} \\ (z,z') \sim f(\cdot; \theta)}} \big[ -\log \zeta(z'|z, a) \big] \quad (5)$$

Importantly, the log-likelihood of a Gaussian distribution with unit variance is equivalent to the mean squared error, differing only by a constant (Hafner et al., 2019). Note that we do not consider training a parameterized reward model because GCRL can leverage intrinsic goal-achieving rewards, akin to reward-free setting (Eysenbach et al., 2022; Park et al., 2023a).

### 4.3. Model-based Policy Training

Model-based policy training is two-fold: 1) data augmentation with a learned dynamic model in Section 4.2 and 2) policy extraction via any existing offline RL algorithm. This process runs iteratively until offline policy converges.

The first step generates representation-action tuples $(z, \widehat{a}, \widehat{z}')$ via model rollouts, where $\widehat{a}$ and $\widehat{z}'$ are getting from a policy $\pi$ and $\zeta$. Generated transition samples are decoded to

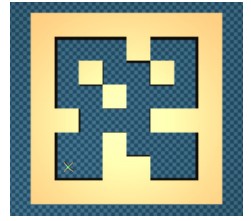 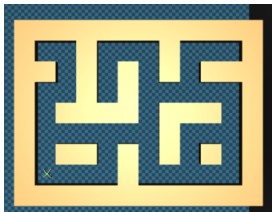 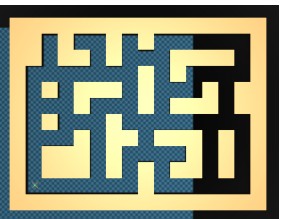 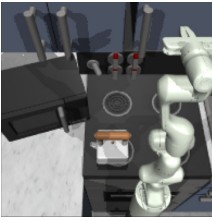 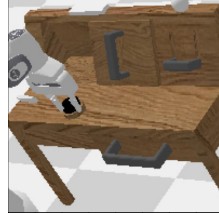

| (a) AntMaze Medium | (b) Antmaze Large | (c) AntMaze Ultra | (d) FrankaKitchen | (e) Calvin |

*Figure 5.* **Selected experimental environments.** (a-c) State-based, single goal-reaching, and long-horizon navigation. (d-e) State-based, multi-goal subtasks, and long-horizon manipulation. We reuse the Kitchen environment as a Pixel-based task.

---

**Algorithm 1** TempDATA with offline RL

---

**Require:** offline dataset $\mathcal{D}$, rollout buffer $\widehat{\mathcal{D}}$, batch sizes $B_\theta, B_\zeta, B_\phi$, learning rates $\eta_\theta, \eta_\zeta, \eta_\phi$, rollout frequency $I$, sampling ratio $\sigma_{\widehat{\mathcal{D}}}$

**Initialize:** network parameters $\theta, \zeta, \phi$, rollout buffer, $\widehat{\mathcal{D}}$, learning rates $\eta_\theta, \eta_\zeta, \eta_\phi$

    Label goal state: $\mathcal{D} \leftarrow \text{GoalLabeling}(\mathcal{D})$
    # Train autoencoder $f_\theta, h_\theta$
    **while** not converged **do**
        Sample $B_\theta$ transitions $(s, s', s_{\text{goal}}) \sim \mathcal{D}$
        $\theta \leftarrow \theta - \eta_\theta \nabla_\theta \mathcal{L}(\theta)$ with $(s, s', s_{\text{goal}})$    # equation (4)
    **end while**
    # Train latent dynamic model $\zeta$
    **while** not converged **do**
        Sample $B_\zeta$ transitions $(s, s', s_{\text{goal}}) \sim \mathcal{D}$
        Map state into representation $(z, z', z_{\text{goal}})$ with $f_\theta$
        $\zeta \leftarrow \zeta - \eta_\zeta \nabla_\zeta \mathcal{L}(\zeta)$ with $(z, a, z')$    # equation (5)
    **end while**
    # Extract offline policy $\pi_\phi$
    **while** not converged **do**
        **if** Iterations **mod** $I == 0$ **then**
            Collect $\widehat{\mathcal{D}} \leftarrow Rollout(\mathcal{D}, \pi_\phi, f_\theta, h_\theta, \zeta)$
        **end if**
        Sample $(1 - \sigma_{\widehat{\mathcal{D}}}) B_\phi$ from $\mathcal{D}$ and $\sigma_{\widehat{\mathcal{D}}} B_\phi$ from $\widehat{\mathcal{D}}$
        Calculate intrinsic reward on $B_\phi$    # equation (6)
        Extract policy with any existing offline RL algorithm
        $\pi_\phi \leftarrow \pi_\phi - \eta_\phi \nabla_\phi \mathcal{L}(\phi)$
    **end while**

---

$(s, \widehat{a}, \widehat{s}')$ using $h(z; \theta)$, and then decoded samples are stored in the augmented rollout dataset $\widehat{\mathcal{D}}$.

Next, in our framework, policy optimization can be divided into two folds: actor-critic methods and weighted SL. Actor-critic methods include a training process of the value function, but we consider a reward-free dataset. Therefore, we define an intrinsic reward to learn goal-conditioned value function efficiently, as follows.

$$\tilde{r}(s, s') = d\big(f(s'; \theta), f(s_{\text{goal}}; \theta)\big) - d\big(f(s; \theta), f(s_{\text{goal}}; \theta)\big) \tag{6}$$

This can be interpreted as an advantage of $s'$ compared to $s$ for goal-achieving. We can train a value function with the following loss function.

$$\mathcal{L}(\phi_Q) = \mathbb{E}_{(s,a,s')\sim\mathcal{D}}\Big[\big(\tilde{r} + \gamma Q(s', \pi(s')) - Q(s, a)\big)^2\Big]$$

Consequently, the weighted SL method can extract a policy without additional value function training, as follows:

$$\mathcal{L}(\phi_\pi) = \mathbb{E}_{(s,a,s')\sim\mathcal{D}}\Big[-\exp(\beta \times \tilde{r}(s, s')) \log \pi(a|s)\Big], \tag{7}$$

where $\beta$ is an inverse temperature (Nair et al., 2020), and $\tilde{r}(s, s')$ can be replaced for an advantage function of the weighted SL (Kostrikov et al., 2022).

### 4.4. Algorithm Summary

Algorithm 1 presents the pseudocode of **TempDATA** comprising three phases. Before beginning the network training, we label the goal information across each transition sample of the dataset using GoalLabeling procedure. We pre-train an autoencoder $\theta$ with an equation (4), and then training latent dynamic model $\zeta$ with an equation (5) and the encoder $f_\theta$. Once the pre-training autoencoder and latent dynamic model are completed, we extract an offline policy based on the offline MBRL solution. We collect rollout data $\widehat{\mathcal{D}}$ using Rollout procedure at each preset rollout frequency $I$. Subsequently, we sample data from two datasets with preset ratios $\sigma_{\widehat{\mathcal{D}}}$, and extract a policy $\pi_\phi$ using off-the-shelf offline RL algorithms. Each phase is reiterated until it converges or during pre-defined iterations. We provide full training details in Appendix A.

> **Remark: Modularity of Temporal Representation**
>
> Our autoencoder is fully compatible with any off-the-shelf model-free or model-based RL algorithm. Moreover, its latent dynamics support a wide array of downstream objectives, *including* RL, skill RL, and GCRL, without requiring any modifications to the core encoding or planning modules.

*Table 1.* Evaluating **TempDATA** (**Proposed**) on D4RL AntMaze environment. The best performance is highlighted in **Bold**.

| AntMaze Dataset | TARL-based methods | | | MBRL-based methods | | | | | |
|---|---|---|---|---|---|---|---|---|---|
| | **S4RL**[†] | **SynthER**[†] | **GTA**[†] | **MOPO**[†] | **RepB-SDE**[†] | **COMBO**[†] | **RAMBO**[†] | **ROMI**[†] | **Proposed** |
| umaze | 55.00±21.0 | 17.1±12.9 | 66.5±13.8 | 0.0 | 0.0 | 80.3±18.5 | 25.0±12.0 | 68.7±2.7 | **96.3**±0.0 |
| umaze-diverse | 51.6±23.4 | 23.9±23.6 | 57.9±19.0 | 0.0 | 0.0 | 57.3±33.6 | 0.0 | 61.2±3.3 | **90.4**±8.6 |
| medium-play | 80.9±10.4 | 41.0±41.2 | **81.9**±8.4 | 0.0 | 0.0 | 0.0 | 16.4±17.9 | 35.3±1.3 | 74.8±8.3 |
| medium-diverse | 74.0±19.4 | 40.1±28.4 | **78.1**±15.8 | 0.0 | 0.0 | 0.0 | 23.2±14.2 | 27.3±3.9 | 69.5±10.8 |
| large-play | 42.9±17.4 | 37.5±13.0 | 44.4±9.3 | 0.0 | 0.0 | 0.0 | 0.0 | 20.2±14.8 | **56.5**±14.1 |
| large-diverse | 46.1±16.7 | 37.5±16.7 | 47.8±13.4 | 0.0 | 0.0 | 0.0 | 2.4±3.3 | 41.2±4.2 | **44.2**±15.3 |
| ultra-play | – | – | – | – | – | 0.0±0.0 | 0.0±0.0 | 4.9±2.1 | **53.2**±18.2 |
| ultra-diverse | – | – | – | – | – | 0.0±0.0 | 0.0±0.0 | 8.8±6.8 | **35.3**±10.9 |
| **Total score** w/o ultra | 350.5 | 236.2 | 376.5 | 0.0 | 0.0 | 137.6 | 88.6 | 253.9 | **431.7** |
| **Total score** | – | – | – | – | – | 137.6 | 88.6 | 272.6 | **520.2** |

*Figure 6.* **Further investigation on performance.** (a) Learning curve about success rate on AntMaze Ultra task. (b) Wall-clock time for each MBRL implementation. (c) RLiable plots (Agarwal et al., 2021) for D4RL benchmark.

# 5. Experiments

In our experiments, we evaluate TempDATA's performance on diverse downstream tasks. In particular, the main performance comparison is performed on three goal-achieving tasks. We also assess the generalizability of TempDATA on pixel-based and dense reward tasks.

**Demonstration Tasks.** We initially outline the environments used for our evaluation, visualized in *Figure 5*. **AntMaze** is a widely-used benchmark environment (Brockman et al., 2016), where an 8-DoF Ant robot navigates to reach a given goal state from the initial one. We consider four different levels of this environment (*i.e.*, Umaze, Medium, Large, and Ultra) and its dataset from D4RL benchmark (Fu et al., 2020). **Kitchen** is a realistic long-horizon benchmark environment (Gupta et al., 2019), where a 9-DoF Franka robot manipulates four different sub-tasks (*i.e.*, open a drawer, move a kettle, etc.). We also use the D4RL benchmark dataset, and we consider two classes ('partial' and 'mixed' without a complete dataset). **CALVIN** another environment designed for long-horizon manipulation tasks (Mees et al., 2022), includes four target subtasks (*i.e.*, push a button, pull the lever, etc.) akin to FrankaKitchen. The main difference is the dataset: CALVIN provides significantly larger, encompassing task-agnostic trajectories from 34 distinct subtasks (Shi et al., 2022; Park et al., 2023a). **Visual Kitchen** is a pixel-based version of Kitchen task.

For dense reward tasks, we adopt four dataset from D4RL benchmark, *including* halfcheetah and walker2D tasks ({"medium-replay" and "medium-expert"}).

**Baseline Algorithms.** We compare the performance of the proposed solution with five offline MBRLs, five offline GCRLs, and three trajectory augmentation-based RL (TARL) methods. For **offline MBRL** methods, we consider model-based policy optimization (MOPO) (Yu et al., 2020), conservative MOPO (COMBO) (Yu et al., 2021) that combines offline MBRL with conservative $Q$ regularizer, robust adversarial MOPO (RAMBO) (Rigter et al., 2022) that trains an adversarial dynamic model against an offline policy, representation balancing with stationary distribution estimation (RepB-SDE) (Lee et al., 2021), and reverse offline model-based imagination (ROMI) (Wang et al., 2021) that trains a reverse dynamic model instead of a forward dynamic model. Next, for **offline GCRL** methods, we use a flat or goal-conditioned variant of SL (GCSL) (Ghosh et al., 2021), CQL (GC-CQL) (Kumar et al., 2020), IQL (or GC-IQL) (Kostrikov et al., 2022), policy-guided offline RL (GC-POR) (Xu et al., 2022) that includes a hierarchy policy structure, and hierarchical IQL (HIQL) (Park et al., 2023a) that is a hierarchical version of GC-IQL. Finally, for **offline TARL** methods, we use S4RL (Sinha et al., 2022), SynthER (Lu et al., 2023), and GTA (Lee et al., 2024c), which are based on the diffusion model.

In the following subsections, our experiments are based on 8 random seeds with two standard deviations (in *Table*) or confidence intervals (in *Figure*). The recorded scores are normalized scores of average success rate on 50 test trials. The performance marked † implies reported benchmark scores by Wang et al.; Rigter et al.; Park et al.; Lee et al..

*Table 2.* Evaluating **TempDATA** on multi-goal tasks, FrankaKitchen and CALVIN.

| Dataset | GCSL[†] | GC-CQL[†] | GC-IQL[†] | GC-POR[†] | HIQL[†] | Proposed |
|---|---|---|---|---|---|---|
| kitchen-mixed | $46.7_{\pm20.1}$ | $15.7_{\pm17.6}$ | $51.3_{\pm12.8}$ | $27.9_{\pm17.9}$ | $\mathbf{67.7}_{\pm6.8}$ | $65.3_{\pm11.7}$ |
| kitchen-partial | $38.5_{\pm11.8}$ | $31.2_{\pm16.6}$ | $39.2_{\pm13.5}$ | $18.4_{\pm14.3}$ | $65.0_{\pm9.2}$ | $\mathbf{70.0}_{\pm12.8}$ |
| CALVIN | $17.3_{\pm14.8}$ | $5.9_{\pm12.3}$ | $7.8_{\pm17.6}$ | $12.4_{\pm18.6}$ | $43.8_{\pm39.5}$ | $\mathbf{50.4}_{\pm34.6}$ |
| **Total score** | 102.5 | 52.8 | 98.3 | 58.7 | 176.5 | **185.7** |

## 5.1. Performance Comparison with MBRL and TARL

This subsection compares the performance of TempDATA with our baseline methods on D4RL AntMaze.

As shown in *Table* 1 (*Left*), TempDATA significantly outperforms the existing offline MBRL methods, especially achieving the best score of 8 out of 8 tasks. TempDATA records a total score difference of approximately 200 compared to the second-best method, ROMI. What is surprising is that TempDATA achieves this even though it does not consider the ensemble structure of the dynamic model, unlike prior MARL methods, and thus can drastically reduce the training time (*Figure* 6(b)). We conjecture this performance improvement comes from model-based additional data generated within the temporal distance-aware representation space. Next, *Figure* 6(a) confirms that TempDATA has comparable or superior performance even when compared with MFRL methods on an extreme-level task. Finally, *Table* 1 and *Figure* 6(c) also demonstrates that the proposed solution mostly achieves the best performance compared to TARL-based methods on D4RL benchmark datasets (AntMaze and Kitchen). Notably, TempDATA attains a substantial performance enhancement of 90.79% over MBRL methods and of 14.66% over TARL ones.

## 5.2. Performance Comparison with GCRL

Next, this subsection compares the performance of TempDATA with GCRL baseline methods on multi-goal tasks from FrankaKitchen and Calvin. As summarized in *Table* 2, TempDATA achieves the highest overall score, outperforming GCRL baseline methods. While the prior model-based approaches did not explicitly target these environments, TempDATA demonstrates that a carefully designed model-based algorithm can rival and even surpass model-free RL techniques on multi-goal tasks.

Notably, TempDATA's advantage is consistent across both the kitchen-mixed and kitchen-partial settings, where it nearly closes the performance gap with HIQL in kitchen-mixed and substantially exceeds it on kitchen-partial. Moreover, on the challenging calvin dataset, TempDATA again records a higher average return compared to all GCRL baselines. These results highlight the flexibility of TempDATA's temporal distance-aware representation, which enables the model to adapt effectively to complex goal-conditioned environments that have not been thoroughly addressed by previous model-based RL approaches.

## 5.3. Performance on Pixel-based Tasks

*Table 3.* Evaluating **TempDATA** on pixel-based task, Visual FrankaKitchen, using two different datasets.

| Visual dataset | GC-IQL[†] | RepB-SDE | Proposed |
|---|---|---|---|
| kitchen-mixed | $52.9_{\pm4.7}$ | $0.00_{\pm0.0}$ | $\mathbf{58.1}_{\pm5.9}$ |
| kitchen-partial | $\mathbf{63.6}_{\pm4.2}$ | $0.00_{\pm0.0}$ | $56.5_{\pm3.5}$ |
| **Total score** | **116.5** | 0.00 | 114.6 |

Lastly, we next evaluate the effectiveness of TempDATA in pixel-based tasks, comparing it against the GCRL baseline (GC-IQL) and an MBRL baseline (RepB-SDE). As shown in *Table* 3, TempDATA achieves performance comparable to the GC-IQL, while RepB-SDE struggles entirely in these pixel-based settings. Surprisingly, this marks a significant improvement for offline MBRL on pixel-based tasks, demonstrating that our proposed solution can successfully overcome the challenges posed by high-dimensional inputs and pave the way for future advances in visual control.

## 5.4. Generalizability on Dense Reward Tasks

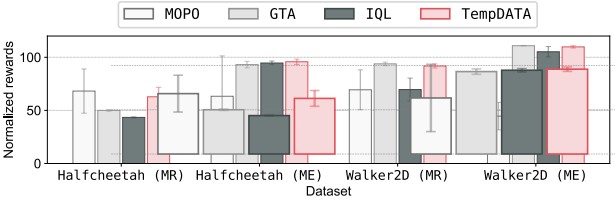

*Figure 7.* **Performance comparison on dense reward tasks of D4RL.** We evaluate the TempDATA on dense reward tasks, *i.e.*, Halfcheetah and Walker2D. In $x$-axes of this plot, MR and ME denote medium-replay and medium-expert datasets.

While our primary focus is on long-horizon, sparse-reward problems, we also evaluated TempDATA on dense-reward benchmarks from D4RL to demonstrate its broader applicability. In *Figure* 7, TempDATA matches or outperforms MOPO, GTA, and IQL across HalfCheetah (MR/ME) and Walker2D (MR/ME), despite not being tailored to dense rewards. These results show that TempDATA scales effectively to classic continuous-control tasks. Comprehensively, these findings validate that the performance gains arise directly from our augmentation framework and underscore the generalizability of TempDATA across diverse RL tasks.

## 5.5. Ablation Study

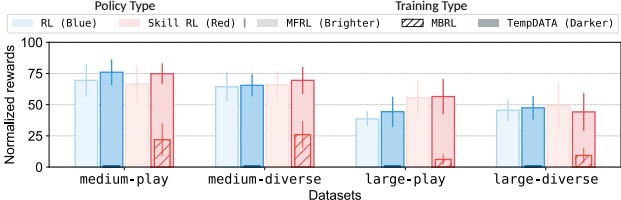

*Figure 8.* **Ablation study by policy and training type on four dataset variants (medium-play, medium-diverse, large-play, large-diverse).** Bright bars (blue: vanilla RL; red: Skill-based RL) show baselines, while darker bars add TempDATA augmentation. Boxed markers underneath indicate naive model-based rollouts.

*Figure* 8 presents an ablation study comparing four algorithmic variants on D4RL datasets. The naive model-based extensions consistently reduce average returns, whereas TempDATA maintains or boosts performance relative to each baseline. In particular, MFRL with naive rollouts suffers a clear drop, but the proposed solution elevates results to match or exceed pure MFRL. Similarly, skill RL alone boosts performance, yet naive rollouts nullify this gain, while the integrated augmentation preserves and enhances the skill RL benefits. This ablation underscores that TempDATA performs better than naive MBRL solutions regardless of policy types.

## 5.6. Scalability on Arbitrary Goal

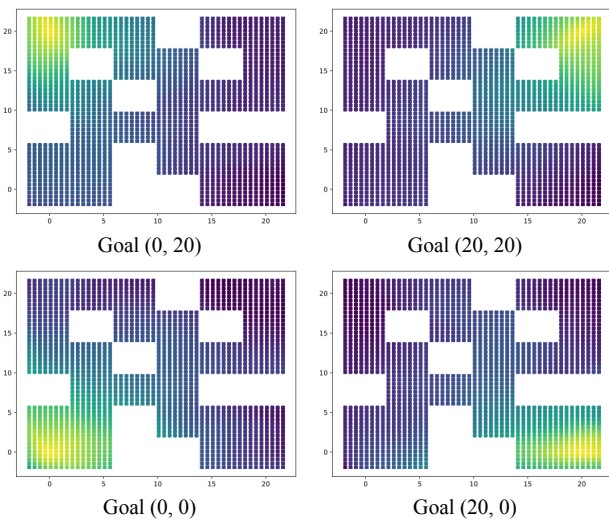

*Figure 9.* **Heatmap according to four different goals in learned latent space.** This visualization shows the distance between every quantized state and goal positions in the learned latent space. Brighter and darker colors imply lower and higher costs.

*Figure* 9 presents heatmap visualizations of learned latent distances between every quantized state and four distinct goal positions at $(0, 20), (20, 20), (0, 0)$, and $(20, 0)$. In each visualization, the heatmap of the representation dis-

tance around obstacles rather than following straight-line Euclidean paths, reflecting the true temporal cost of navigation. Notably, the shape and spacing of these contours closely match the shortest-path travel times to each goal, indicating that the encoder has internalized environment dynamics. This consistency across all four goal tasks shows that the autoencoder captures a universal temporal metric, not merely goal-specific shortcuts. Consequently, given an arbitrary goal at a test time, TempDATA can approximate the true effort, as a temporal distance, required to reach that goal. These quantitative heatmaps confirm that TempDATA generalizes robustly to unseen goal locations by aligning learned distances with obstacle-aware travel times. Together with our ablation studies, this evidence demonstrates that a single learned representation suffices to plan efficient paths for varied goal-reaching scenarios.

# 6. Conclusion

This work introduces a novel offline MBRL method, TempDATA, which augments new transitions in a latent space instead of a raw state space. This solution builds a representation space that should capture the temporal distance between microscopic and macroscopic levels. Next, we train a latent dynamic model to generate new transitions, thereby leveraging off-the-shelf offline RL algorithms. The proposed solution not only outperforms the offline MBRL approaches in challenging goal-reaching benchmarks but also competes favorably with the GCRL or TARL approaches.

**Closing Statements.** The combination of model-based RL and temporal encoding in latent space distills a fixed dataset into a compact latent space that encodes both the dynamics and the geometry needed for long-horizon planning and robust policy extraction. Although we found such efficiency and robustness, there are still questionable points.

- How well do these model-based rollouts venture beyond the empirical support?

- What if observations are partial, noisy, or stochastic, and global distance preservation shatters?

- How can an ego-agent embed worlds where other agents, not just itself, drive state transitions?

- Why restrict ourselves to symmetric metrics when many tasks are temporally asymmetric?

- Should the offline pre-training remain strictly offline?

Answering these questions will chart the course for truly generalizable, distance-aware control: by blending robust representation learning with principled uncertainty modeling, multi-agent reasoning, asymmetric geometry, and hybrid online–offline adaptation, we can move beyond today's safe but limited rollouts toward methods that excel in real-world, long-horizon decision making.

## Impact Statement

This paper presents TempDATA, a general method for synthesizing temporal-distance-aware transitions that guide goal-reaching policies without extra environment interactions. By generating meaningful data in latent space, our approach accelerates learning of long-horizon behaviors and reduces reliance on costly real-world trials. TempDATA's framework can broadly enhance sample efficiency and robustness in offline RL applications.

## Acknowledgement

We thank the ICML 2025 reviewers for their constructive feedback. We also appreciate Prof. Younghan Kim, who is affiliated with Soongsil University, for his computing resource support for some parts of the experiments. This work was supported by by Institute of Information & communications Technology Planning & Evaluation (IITP) grant funded by the Korea government (MSIT) (RS-2022-00143911, AI Excellence Global Innovative Leader Education Program), NRF grant (RS-2023-00278812), and in part by the IITP grants (No. 2021-0-00739, IITP-2025-RS-2020-II201602) funded by the Korea government (MSIT).

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

# A. Training Details

**Convolutional encoder for the pixel-based task.** For pixel-based environments, we handle image input using the IMPALA CNN architecture (Espeholt et al., 2018). The TempDATA auto-encoder uses the 512-dimensional output features of Impala CNN. In addition to the auto-encoder, our dynamic model and RL policy also use such compressed image features of IMPALA CNN. Therefore, we do not need to build a raw image decoder model.

**Goal-conditioned Reinforcement Learning.** We train a skill-conditioned policy in an unsupervised manner to enable efficient learning in goal-achieving environmental settings. Moreover, this solution can be applied to multiple goal scenarios instead of fixed goal scenarios. We introduce a latent skill variable $w$, which is randomly sampled from $p(w)$ inspired by DIYAN, and learn a policy $\pi(a|s, w)$ to explore and capture diverse behaviors from a reward-free offline dataset. This unsupervised phase leverages a pre-trained encoder to compute rewards and measure skill progress in the latent space. Specifically, as we discussed in Section 4.3, we define an intrinsic reward, $\tilde{r}(s, s') = d\big(f(s'; \theta), f(s_{\text{goal}}; \theta)\big) - d\big(f(s; \theta), f(s_{\text{goal}}; \theta)\big)$. Next, we implement our offline RL training using the IQL combined with AWR objective function, ensuring stable policy improvement in an in-sample learning manner. Our overall pipeline is integrated within the TempDATA codebase, which extends HIQL implementations (Park et al., 2023a). However, we do not consider a hierarchical policy structure in this work.

**Policy execution.** At the test phase, the trained policy requires a specific skill variable according to the given goal state. To do this, we adopt the test-time skill adaptation, which is proposed in HILP (Park et al., 2024), as follows:

$$w^* = \frac{f(s_{\text{goal}}; \theta) - f(s; \theta)}{\|f(s_{\text{goal}}; \theta) - f(s; \theta)\|},$$

which guides the policy to decrease the distance to $f(s_{\text{goal}}; \theta)$ by moving the latent direction $f(s_{\text{goal}}; \theta) - f(s; \theta)$. This approach requires no additional training and can be performed in a zero-shot manner.

**Procedures: Goal labeling.** To train autoencoder and offline RL networks, we relabel goal distributions using hindsight experience relabeling (Andrychowicz et al., 2017). Specifically, for a randomly selected state, we assign it as the goal with a $20\%$ probability, a future state from the trajectory with a $50\%$ probability, and a completely random state with a $30\%$ probability. The reward is set to $-1$ for every timestep until the goal is achieved, at which point the done mask is set to True. During the encoding process, we ensure that at least one of the samples contains the goal state.

**Procedures: Model rollout.** We adopt a standard model rollout approach (Yu et al., 2020) in which a fully trained dynamics model is used to simulate additional trajectories during policy learning. Specifically, the agent performs rollouts of length $k$ steps in the learned model to generate synthetic state-action-reward transitions that complement the original offline dataset. To ensure a stable and well-grounded initialization, we train the policy for the first $30\%$ of the total training duration without any rollouts. This phase provides the agent with a direct understanding of the environment's dynamics before synthetic samples are introduced. Subsequently, at every $10\%$ iteration of the remaining training period, we generate synthetic samples equivalent to half the current size of the offline buffer and add them to the training pool. By gradually incorporating model-generated data, we allow the agent to refine its decision-making while mitigating potential inaccuracies from extensive model rollouts.

# B. Proof of Theorem 4.2

*Proof.* We use a Bellman operator with expectile regression to learn $d_\theta(s, s_{\text{goal}})$. Concretely, each update enforces

$$d\big(f(s; \theta), f(s_{\text{goal}}; \theta)\big) \approx 1 + \gamma \min_a d\big(f(s'; \theta), f(s_{\text{goal}}; \theta)\big),$$

where $\gamma$ is the discount factor, and $(s, a, s')$ are transitions sampled from the dataset or replay buffer. The loss itself is formulated via expectile regression, which replaces the usual squared Bellman error with a $\tau$-expectile operator.

A property of expectile regression states that if $X$ is a bounded random variable with supremum $x^*$, then

$$\lim_{\tau \to 1} m_\tau(X) = x^*,$$

where $m_\tau(X)$ is the $\tau$-expectile of $X$. Translating this to our Bellman setting, driving $\tau$ toward 1 forces the learned distance $d_\tau$ to match the supremum of returns along feasible trajectories.

In a deterministic environment, the total cost from $s$ to $s_{\text{goal}}$ along any path is at least as large as the *shortest-path* cost. From standard arguments, we have:

$$d_\theta(s, s_{\text{goal}}) \leq (\text{shortest-path cost from } s \text{ to } s_{\text{goal}}) = -V^*(s, s_{\text{goal}}).$$

Meanwhile, the Bellman-like constraints in our training also ensure that $d_\theta$ cannot collapse below the maximum relevant cost, since the expectile objective pushes it toward the worst-case outcome as $\tau \to 1$.

Because $d_\theta(s, s_{\text{goal}})$ is bounded both above and below by the same shortest-path or $-V^*$ cost in the limit, it converges exactly to

$$d_\theta(s, s_{\text{goal}}) = -V^*(s, s_{\text{goal}}).$$

Equivalently,

$$\lim_{\tau \to 1} d_\tau\big(f(s; \theta), f(s_{\text{goal}}; \theta)\big) = -\max_\pi V^\pi(s, s_{\text{goal}}),$$

completing the proof. $\square$

## C. Implementation Details

Our implementation for TempDATA is based on JaxRL and is available at the following repository. We run our experiments on RTX 3090 GPUs. Each experiment runs for no more than 3 hours in state-based environments and no more than 12 hours in pixel-based environments.

### C.1. Environments

**AntMaze.** We examine eight distinct scenarios within the AntMaze task: 'antmaze-umaze-v2', 'antmaze-umaze-diverse-v2', 'antmaze-medium-diverse, play-v2', 'antmaze-large-diverse, play-v2', and 'antmaze-ultra-diverse, play-v0'. The datasets for the 'umaze', 'medium', and 'large' scale environments originate from the D4RL benchmark (Fu et al., 2020), while the datasets for AntMaze-ultra are sourced from a separate work. Notably, the AntMaze-ultra environment (Jiang et al., 2022) is twice as large as AntMaze-large. Each dataset comprises 999 trajectories, each with a length of 1000 steps, where the Ant agent moves from a randomly chosen starting position to a goal location, which is not necessarily the evaluation target. During testing, to define a goal $g$ for the policy, we modify the first two state dimensions—corresponding to x-y coordinates—to the designated target position in the environment, while the remaining proprioceptive state variables are set to those from the first observation in the dataset. The agent receives a reward of 1 upon successfully reaching the goal during evaluation.

**Kitchen.** The Kitchen datasets from D4RL (Fu et al., 2020) consists of two datasets as 'kitchen-{partial, mixed}-v0'. These datasets capture trajectories of a robotic arm interacting with various objects in different sequences within its environment. For this task, the agent gets a reward of 1 upon completing each subtask, with each episode comprising four subtasks. In addition to state-based work, for pixel-based Kitchen experiments, we transform each state into a $64 \times 64 \times 3$ camera image through rendering, employing the same camera configuration used by Mendonca et al. and Park et al..

**CALVIN.** The CALVIN offline datasets are rooted in the teleoperated demonstrations by Mees et al. and, thereby being introduced by Shi et al. and Park et al.. This dataset comprises 499 trajectories; each trajectory has 1204 transitions, capturing various subtasks performed in an arbitrary order. Similar to the FrankaKitchen task, the CALVIN task consists of four subtasks and the agent obtains a reward of 1 upon completing each subtask.

### C.2. Hyperparameters

| Hyperparameter | Value |
|---|---|
| Iterations | $10^6$ (state-based), $5 \times 10^5$ (pixel-based) |
| Learning rate | $3 \times 10^{-4}$ (all networks) |
| Optimizer | Adam (Diederik, 2015) |
| Batch size | 512 (AntMaze), 256 (FrankaKitchen), 128 (CALVIN) |
| The number of evaluation episodes | 50 (all tasks) |
| Dimensions for autoencoder network | $[512, 512, 512, \{32, 10\}, 512, 512, 512]$ 
 32 (AntMaze), 10 (CALVIN, FrankaKitchen) |
| Discount factor for autoencoder | 0.99 (all tasks) |
| Expectile coefficient for Autoencoder | 0.95 (AntMaze), 0.97 (CALVIN, FrankaKitchen), 0.7 (Pixel-based) |
| Dimensions for dynamic model network | $[512, 512, 512]$ |
| The number of rollout steps | 3 |
| Dimensions for critic network | $[512, 512, 512]$ |
| Dimensions for actor network | $[512, 512, 512]$ |
| Target smoothing coefficient | $5 \times 10^{-3}$ |
| Discount factor for offline RL | 0.99 |
| Inverse temperature for offline RL | 10 (AntMaze), 3 (CALVIN, FrankaKitchen) |
| Expectile coefficient for offline RL | 0.9 (state-based), 0.7 (pixel-based) |

