# OpenReview forum: "Temporal Distance-aware Transition Augmentation for Offline Model-based Reinforcement Learning"
_ICML.cc/2025/Conference — ICML 2025 poster_

### Official Review · Reviewer_rU4d · 2025-03-09

**Overall Recommendation:** 3

**Summary:**

This paper focuses on the failure of model-based reinforcement learning (MBRL) in sparse-reward and long-horizon environments, emphasizing that the key to addressing this issue lies in generating data that incorporates temporal information. To tackle this challenge, this paper introduces a novel MBRL framework, TempDATA, which consists of three components: an autoencoder, a latent dynamic model, and an offline policy. The autoencoder is trained to learn state abstractions that capture temporal distances at both the trajectory and transition levels in a representative space and then reconstruct these abstractions in the original state space. The latent dynamic model is trained to augment the dataset with temporal distance-aware transition, while the offline policy is extracted from the augmented dataset.

## update after rebuttal

Thank the authors for the detailed response, and I have reviewed the additional comments. The results of these two experiments are truly impressive. The first experiment provides excellent insights into the alignment between distances in the encoded representation space and obstacle-induced temporal distances, while the second experiment further offers quantitative evidence to support this point. I have no additional questions. Given the innovation and thoroughness of the paper, I believe a score of 3 is appropriate, and I will maintain this rating.

**Claims And Evidence:**

The claims made in the paper are supported by clear evidence.

**Essential References Not Discussed:**

No

**Experimental Designs Or Analyses:**

There’s a doubt about the experiment. The TempDATA is tested in four datasets: AntMaze, Kitchen, Calvin, and Visual Kitchen with thirteen baselines. However, some baselines were only tested for part of the four datasets. I suppose that the reason for this experimental design isn’t mentioned in the paper. Additionally, while I acknowledge the superior performance of TempDATA across various benchmarks, the paper would be even stronger if additional experiments were included to demonstrate the autoencoder’s ability to capture temporal information.

**Methods And Evaluation Criteria:**

It makes sense for problem.

**Other Comments Or Suggestions:**

No

**Other Strengths And Weaknesses:**

I appreciate the effort put into providing an intuitive understanding of TempDATA. These illustrative figures were helpful in clarifying the key points of the paper. However, the writing of this paper could be improved. Spelling errors and grammatical mistakes hinder the reading experience and make some of the claims unclear at first glance. Therefore, the author should revise the paper to enhance its clarity.

**Questions For Authors:**

See the experimental designs or analyses part.

**Relation To Broader Scientific Literature:**

The paper is well-grounded in the broader literature.

**Theoretical Claims:**

I’ve checked the correctness of the theorem proof mentioned in the main paper.

---

> ### Author Rebuttal · Authors · 2025-03-31
>
> We thank the reviewer for the positive reviews and insightful feedback about this work. Below, we describe how we have revised the paper to address the reviewer's concerns and questions.
>
> ---
> # Experimental coverage of baseline algorithms
> We thank the reviewer for pointing this out. In our work, we designed the experimental evaluation around three types of tasks:
> 1) **AntMaze**, which is a single-goal long-horizon task
> 2) **CALVIN and FrankaKitchen**, which require sequential multi-goal manipulation (sub-goal decomposition tasks)
> 3) **Visual FrankaKitchen**, which is based on image-rendered dataset version of FrankaKitchen (pixel-based goal-conditioned tasks)
>
> We mainly compare with offline MBRL and TARL baselines on the AntMaze task, where such model-based approaches have been previously studied extensively. However, for the sub-goal decomposition tasks (e.g., CALVIN and FrankaKitchen) and pixel-based tasks, **prior offline MBRL and TARL methods have not been explored** in the existing literature.
>
> To investigate their applicability further, we ran above-mentioned experiments by applying representative offline MBRL and TARL baselines to these tasks. However, **we observed that the resulting performance was consistently near-zero**, with success rates failing to exceed random behavior. We attribute this to the severe sparsity of rewards and the difficulty of long-horizon goal decomposition in these environments.
>
> Given these empirical observations, **we concluded that such methods do not serve as meaningful baselines in these domains, and instead chose to compare against goal-conditioned model-free RL (GCRL) methods**, which are more capable of handling these challenges. We will explicitly highlight this reasoning in the revised manuscript to avoid confusion.
>
> ---
> # Evidence of temporal representation
>
> We appreciate the reviewer’s suggestion. We would like to clarify that Figure 3 in our manuscript directly addresses this concern; Reviewer p9eS and DPTt agree that Figure 3 justifies representation encoding qualitatively. The figure provides an empirical visualization of the learned latent space using t-SNE embeddings, which demonstrates that **the trained encoder maps states according to their temporal distance rather than their spatial similarity**.
>
> Specifically, if the encoder fails to capture temporal distance, the t-SNE projection results in an entangled and structureless cloud. In contrast, our result shows a meaningful geometric progression, where temporally distant states are mapped further apart, confirming that the encoder preserves temporal information. We strongly believe that the provided visualization qualitatively validates the temporal property of the embedding. We will explicitly emphasize and clarify this qualitative validation in the revised manuscript to strengthen our claims further.
>
> ---
> Please let us know if there are any additional concerns or questions.

---

> > ### Comment · Reviewer_rU4d · 2025-04-04
> >
> > Thank you for your concrete feedback. The explanation in the Experimental coverage of baseline algorithms part is convincing and greatly solved my confusion about the experimental setting before. For the response in the Evidence of temporal representation part, I acknowledge that Figure 3 provides a great intuition about the superior ability of the trained encoder in representation encoding. However, I insist that supplementing an experiment quantifying the temporal distance between the encoded representation could be an option to enhance the soundness of the paper, since the encoder is one of the main contributions of this paper.
> >
> > ---
> > Thank you for the detailed response, and I have reviewed the additional comments. The results of these two experiments are truly impressive. The first experiment provides excellent insights into the alignment between distances in the encoded representation space and obstacle-induced temporal distances, while the second experiment further offers quantitative evidence to support this point. I have no additional questions. Given the innovation and thoroughness of the paper, I believe a score of 3 is appropriate, and I will maintain this rating.

---

> > > ### Author Response · Authors · 2025-04-05
> > >
> > > We greatly appreciate your constructive suggestions.
> > >
> > > ---
> > > We have clearly understood what the reviewer wants, so we have performed additional experiments to supplement our qualitative results with clear quantitative evidence.
> > >
> > > First, to quantitatively illustrate how effectively our encoder captures temporal information, we have provided additional heatmap visualizations mapping the representation distances between goal states and various points across the state space. Specifically, we selected four distinct goal positions located at (0.0, 20.0), (20.0, 20.0), (0.0, 0.0), and (20.0, 0.0), and **visualized how distances in the encoded representation space align with obstacle-induced temporal distances.** These results show the encoder’s sensitivity not only to simple Euclidean proximity but also to obstacle-based navigation constraints.
> > >
> > > We have included these additional experimental results. A detailed figure illustrating these results can be accessed via the following anonymous web link: https://sites.google.com/view/icml-rebuttal-1959/home
> > >
> > > Moreover, we would like to provide a specific example for an explicit numerical value that quantifies representation distances between selected states. As shown below, our encoder successfully captures meaningful temporal relationships between states:
> > >
> > > |             | (0.0, 20.0) | (4.0, 12.0) | (8.0, 16.0) |
> > > | ----------- | ----------- | ----------- | ----------- |
> > > | (0.0, 20.0) |             | 14.8        | 15.6        |
> > > | (4.0, 12.0) | 14.8        |             | 33.7        |
> > > | (8.0, 16.0) | 15.6        | 33.7        |             |
> > >
> > > Notably, the states at (4.0, 12.0) and (8.0, 16.0) are spatially close in terms of Euclidean distance, yet due to obstacles, they are temporally distant. Our encoder accurately reflects this temporal discrepancy in the encoded representation (distance = 33.7). **This demonstrates clearly that the representation space meaningfully captures nontrivial navigation dynamics and temporal information beyond simple spatial proximity.**
> > >
> > > In the anonymous web, you can check each point's coordinates at the most-below figure.
> > >
> > > Blue: (0.0, 20.0) / Red: (4.0, 12.0) / Orange: ( 8.0, 16.0)
> > >
> > > ---
> > > Thank you again for your insightful comments, which substantially enhanced the clarity and robustness of our analysis and paper. If these additional experiments and explanations have addressed your concerns, we would be grateful if you could consider revising your score accordingly. Thank you once more for your thoughtful consideration.

---

### Official Review · Reviewer_DPTt · 2025-03-13

**Overall Recommendation:** 4

**Summary:**

This paper proposes TempDATA, a new offline model-based reinforcement learning (MBRL) method that learns a temporal-distance-aware autoencoder model, a latent dynamic model, and an offline policy. Specifically, TempDATA first trains the autoencoder and latent dynamic model to possess the temporal-distance attribute. Then, it generates augmented rollouts using the learned autoencoder and latent dynamic model. Lastly, it learns the offline policy by leveraging both original and augmented transitions. The proposed TempDATA is evaluated on tasks from multiple benchmarks, including AntMaze, FrankaKitchen, and CALVIN. From the results, the paper concludes that TempDATA achieves competitive or improved performance compared to various baselines.

**Claims And Evidence:**

The main claim made in this paper is that the policy learns with augmented transitions, generated by the temporal-distance-aware system (autoencoder + latent dynamic models), and can achieve better results.

[+] From the theoretical aspect, the proposition is sound, though it has some confusing parts. From the empirical aspect, the proposed method significantly outperforms baselines in state-based environments, and the visualization result (Figure 3) further supports this claim qualitatively.

[-] However, I believe there is a prerequisite for the paper to obtain this result: the offline dataset itself must cover a sufficient range of transitions to effectively build/approximate the dynamic model. More details will be summarized below.

**Essential References Not Discussed:**

[+] The paper discusses and cites sufficient works. The literature review is well summarized in the related work section. Additionally, the proposed method is compared against baselines of various types and attributes.

**Experimental Designs Or Analyses:**

[+] The results for each environment are tested over multiple rounds, and the standard deviation of the method’s performance is also reported.

[+] TempDATA is compared with several baseline methods, showing a significant performance improvement in state-based environments.

[-] In image-based environments, TempDATA achieves sub-optimal performance, raising concerns about its ability to learn the dynamic model when working with data that contains only partial information—something commonly encountered in practice. The explanation for why TempDATA performs sub-optimally is insufficient. Additional experiments in image-based environments, along with a more detailed analysis, could help address this concern.

[-] As mentioned earlier, I believe the proposed method outperforms MFRL methods because the offline dataset provides enough state-action coverage to learn an effective dynamic model. However, when the offline dataset is small or lacks data in critical areas, I feel that MBRL methods will be significantly impacted. Conducting experiments on this aspect is essential to alleviate this concern.

**Methods And Evaluation Criteria:**

[+] While the components in TempDATA, namely temporal-distance-aware latents and the latent dynamic model, may have been explored individually in previous work, their combination and usage in MBRL make sense to me. The motivation to introduce the temporal-distance-aware autoencoder is also supported by theoretical backing.

[+] The proposed method is evaluated on benchmarks across various domains, such as robot arm and maze navigation. These benchmarks include both state-based and image-based environments.

[-] Some evaluation metrics lack descriptions; for instance, the IQM and Optimality gap in Figure 6 (c).

**Other Comments Or Suggestions:**

All my comments and suggestions are listed in the appropriate fields above. Regarding my recommendation, I actually feel the paper is around the threshold; I would choose "borderline" if there were such an option. Based on the observed pros and cons, I will set my initial score as "weak accept." I would be happy to further adjust it if my concerns are well-addressed or clarified.

---

**Comments after author-reviewer discussion**

Thank you for providing additional experiments on data scarcity. As promised, I will increase my score to "accept." That said, if the paper is accepted, please make sure to incorporate these updates and revisions into the camera-ready version. I also agree with Reviewer p9eS that the writing in the method section—especially the mathematical formulations—can be further improved. For future work, it would be valuable to evaluate the proposed TempDATA on long-horizon tasks and assess whether it still demonstrates effectiveness in estimating the temporal distance to task completion.

**Other Strengths And Weaknesses:**

[+] The training details provided in the supplementary materials improve the reproducibility of this work.

[-] While the paper is generally easy to follow, I found the following points that need correction or revision:
- [line 107 (right)]: "image-based state" → From a rigorous perspective, an image is an observation of a state that only contains partial information, rather than a type of state.
- [line 113 (right)]: "or state-action value function $V(s)$" → Should this be "state value function" ?
- [line 116 (right)]: $\arg  \underset{a' \sim \pi(s')}{\max} \, Q(s', a')$ -> $\underset{a' \sim \pi(s')}{\arg \max} \, Q(s', a')$
- [line 221]: The parameters required by the function $d$ are inconsistent between Eq. 1 and line 221.
- [line 252 (right)]: In the formula, $ds$ and $ds^{'}$ are usually placed at the end, not at the beginning.

**Questions For Authors:**

Please consider addressing the concerns listed in the previous sections. I will highlight some important ones here:

1. How much does the performance of the dynamic model influence the proposed method? Specifically, if the offline dataset is limited or biased, what impact does this have on the method’s effectiveness?
2. What explains the sub-optimal performance of the proposed method in image-based environments?
3. Could you provide further clarification regarding the design of the reward function $r_g(s)$?

**Relation To Broader Scientific Literature:**

[+] In my view, offline RL is a challenging yet valuable problem, especially for real-world applications where data collection is costly, risky, or impractical. Advancing offline RL methods has the potential to significantly impact related fields, such as robotic learning.

**Theoretical Claims:**

[+] As mentioned above, Proposition 4.1 and its proof are sound in general.

[-] It is unclear why the reward function is designed as $r_g(s)-1$. Especially in line 225, it is specifically noted that this design differs from the vanilla goal-conditioned reward function, but the associated footnote indicates, "General methods use transitions by subtracting 1 from $r_g(s)$." Isn't this exactly $r_g(s)-1$? Why is the reward function designed this way? Is there any physical meaning to support it?

[-] While the proposition is sound, I believe that in practice, how well the offline dataset covers the space of possible states and actions will significantly affect the method's performance—i.e., how well the dynamic/world model can predict the observation/state/latent for the next time step.

I believe this is a common challenge within the MBRL approach, which is why I am skeptical of the argument in lines 42-46 (right) of the paper, as quoted: "*The offline MBRL methods have covered OOD samples efficiently, achieving a better performance than offline MFRL in dense rewarded or short-horizon robotic manipulation tasks.*"

If there is no such state-action pair in the offline dataset, it is inherently impossible for the dynamic model to learn and make correct predictions.

---

> ### Author Rebuttal · Authors · 2025-03-31
>
> We appreciate the reviewer's time and effort. Here are our answers to the reviewer's comments.
>
> ---
> # Performance according to dataset
> We agree with the concern raised by the reviewer. To address the concern, we ran additional experiments by gradually reducing the offline dataset coverage for the antmaze-medium tasks.
>
> | | 100%| 80%| 60%| 40%| 20%|
> |-|-|-|-|-|-|
> | antmaze-medium-play| $74.8\pm8.3$  | $78.6\pm10.0$ | $65.5\pm15.5$ | $41.4\pm6.6$ | $11.9\pm5.0$ |
> | antmaze-medium-diverse | $69.5\pm10.8$ | $69.1\pm9.3$  | $49.8\pm18.2$ | $34.8\pm8.2$ | $18.0\pm7.6$ |
>
> While performance drops significantly when the dataset is extremely limited at $20$% coverage, experimental results demonstrate that performance remains relatively robust with up to $60$% coverage. These results indicate that the representation can still be learned effectively under moderate data scarcity. To sum up, TempDATA is robust under moderate dataset scarcity, but there is room for improvement under severe limitations or bias in the dataset.
>
> ---
> # TempDATA with pixel-based tasks
> To verify the competence of TempDATA in the pixel-based RL, we ran additional experiments on a pixel-based benchmark: Visual AntMaze. Visual AntMaze is a variant of AntMaze with camera images and proprioceptive states, requiring the agent to navigate based on visual cues. This dataset is rendered by using D4RL antmaze-medium and -large datasets.
>
> | visual-antmaze dataset | RepB-SDE | GC-IQL| TempDATA|
> |-|-|-|-|
> | medium-play| $0\pm0$  | $58.8\pm12.0$ | $60.8\pm11.6$ |
> | medium-diverse| $0\pm0$  | $65.6\pm19.8$ | $62.0\pm12.2$ |
> | large-play| $0\pm0$  | $35.7\pm10.2$ | $42.8\pm9.7$  |
> | large-diverse| $0\pm0$  | $29.0\pm8.8$  | $40.8\pm7.0$  |
>
> These results confirm that TempDATA effectively handles pixel-based tasks, achieving notable performance compared to GC-IQL and RepB-SDE. **Despite challenges inherent in augmenting visual data augmentation, our solution underscores its competence unlike other MBRL solution**.
>
> ---
> # Reward for skill-conditioned RL
> We appreciate the reviewer’s careful observation regarding the reward design $r_g(s) - 1$, and we acknowledge that our wording causes confusion between the terms ``vanilla`` and ``general``. Specifically, the vanilla reward refers to the binary scenario ($r_g(s) = 1$ for goal states, $r_g(s) = 0$  otherwise). our footnote inadvertently associated this definition with the general method (subtracting $1$ from $r_g(s)$).
>
> To clarify, the vanilla reward yields sparse signals, hindering effective learning of long-horizon tasks. Therefore, we adopted the general alternative $r_g(s) - 1$, as it provides clearer temporal information and better learning signals [1, 2].
>
> In our framework, this transformation plays a crucial role during the pre-training stage, where the agent learns a temporally meaningful value function. By consistently assigning a penalty $(-1)$ for non-goal states and 0 for goal states, **we enable the Q-function to represent a form of temporal distance to the goal, as $Q(s, a) \approx -d(s, g)$, where $d(s, g)$ is the minimum number of steps to reach the goal from state $s$**. This mathematical property allows our Q-function to encode structured temporal knowledge, which is especially valuable in downstream tasks requiring generalization over time and space.
>
> ---
> # Challenge of MBRL
> We fully agree that the coverage of the offline dataset significantly impacts the performance of offline MBRL methods, particularly regarding their ability to generalize to OOD states and actions. However, the original statement aimed at comparing offline MBRL with MFRL methods under offline datasets in dense-rewarded or short-horizon robotic tasks. Even under an identical offline dataset, MFRL methods rely exclusively on observed trajectories and thus inherently lack the inductive biases that enable generalization beyond the seen data. In contrast, offline MBRL can leverage learned dynamics to partially alleviate mild OOD challenges through structured forward prediction. Nevertheless, we acknowledge that this statement was overly optimistic and could be misleading. We will make efforts to tone down the over-claim about MBRL.
>
> ---
> Thank you for your helpful comments on the clarity and correctness of our writing. We will revise our manuscript and include additional information, e.g., IQM and optimality Gap.
>
> Please let us know if you have any additional concerns or questions.
>
> # Reference
> [1] A. Kumar, et al. Conservative Q-learning for offline reinforcement learning. NeurIPS 2020.
>
> [2] K. Frans, et al. Unsupervised zero-shot reinforcement learning via functional reward encodings. ICML 2024.

---

> > ### Comment · Reviewer_DPTt · 2025-04-02
> >
> > I appreciate the authors' effort in addressing my concerns, and many of them have been well addressed.
> >
> > One remaining concern is the relationship between data coverage and performance. While I appreciate the additional experimental results, I believe TempDATA's performance should be compared with at least one SOTA model-based RL (MBRL) and one SOTA model-free RL (MFRL) method under the same data coverage setting to better demonstrate its robustness to data scarcity.
> >
> > Regarding my recommendation, I now slightly lean toward the positive side, so I will maintain my 'weak accept' recommendation for now. If the requested experiment is provided and the proposed method demonstrates better robustness against data scarcity, I will be happy to raise my score to 'Accept.'

---

> > > ### Author Response · Authors · 2025-04-05
> > >
> > > We appreciate the reviewer for acknowledging our efforts and for the willingness to engage in further discussion to strengthen our work.
> > >
> > > ---
> > > Following the reviewer's suggestion, we ran additional experiments to compare TempDATA's robustness to data scarcity by comparing its performance against widely used offline MFRL (IQL and CQL) and MBRL (ROMI). The detailed results are presented below:
> > >
> > > | antmaze-medium-play    | 100%          | 80%           | 60%           | 40%           | 20%           |
> > > | ---------------------- | ------------- | ------------- | ------------- | ------------- | ------------- |
> > > | TempDATA               | $74.8\pm8.3$  | $78.6\pm10.0$ | $65.5\pm15.5$ | $41.4\pm6.6$  | $11.9\pm5.0$  |
> > > | IQL (MFRL) | $75.4\pm7.8$  | $70.2\pm5.3$  | $55.1\pm9.9$  | $39.8\pm13.9$ | $13.3\pm7.1$  |
> > > | CQL (MFRL)| $65.5\pm13.2$ | $44.0\pm15.1$ | $11.7\pm10.4$ | $2.7\pm3.1$   | $0.0\pm0.0$   |
> > > | ROMI (MBRL)| $35.3\pm1.3$  | $30.6\pm2.8$  | $21.4\pm2.5$  | $10.9\pm6.5$  | $3.7\pm3.3$   |
> > >
> > > | antmaze-medium-diverse | 100%          | 80%           | 60%           | 40%           | 20%           |
> > > | ---------------------- | ------------- | ------------- | ------------- | ------------- | ------------- |
> > > | TempDATA               | $69.5\pm10.8$ | $69.1\pm9.3$  | $49.8\pm18.2$ | $34.8\pm8.2$  | $18.0\pm7.6$  |
> > > | IQL (MFRL)| $65.0\pm10.2$ | $71.6\pm12.4$ | $52.3\pm9.3$  | $26.8\pm14.0$ | $12.3\pm10.4$ |
> > > | CQL (MFRL)| $50.0±15.7$   | $35.7\pm13.4$ | $24.8\pm14.0$ | $7.3\pm8.5$   | $1.0\pm2.0$   |
> > > | ROMI (MBRL)| $27.0\pm3.5$  | $29.8\pm9.2$  | $10.8\pm7.0$  | $9.6\pm6.4$   | $5.2\pm3.8$   |
> > >
> > > This result shows that TempDATA significantly surpasses the MBRL baseline (ROMI). Compared to CQL, TempDATA consistently outperforms with a significant margin, especially under severe data scarcity. While TempDATA shows similar robustness to the strong MFRL baseline (IQL), the proposed solution demonstrates a slight average advantage across sparsity levels, suggesting more stable performance. The performance degradation of MFRL methods against data sparsity is similar to what was previously reported in [1, 2].
> > >
> > > We believe these additional results address the reviewer’s remaining concern.
> > >
> > > ---
> > > # References
> > > [1] P. Cheng, et. al. Pushing the Limit of Small-Efficient Offline Reinforcement Learning. OpenReview. 2025.
> > >
> > > [2] P. Cheng, et al. Look beneath the surface: Exploiting fundamental symmetry for sample-efficient offline RL. NeurIPS. 2023.
> > >
> > > ---
> > >
> > > **Update Apr 07**: We understand that only one rebuttal reply is allowed for the reviewers. If you have any additional comments, please feel free to update your existing comments above, and we will continue to monitor them. Additionally, if you feel our response has sufficiently addressed your concerns, we would appreciate it if you could kindly consider adjusting your score accordingly.

---

### Official Review · Reviewer_Akpp · 2025-03-14

**Overall Recommendation:** 3

**Summary:**

This paper addresses the challenges of offline model-based reinforcement learning, particularly in sparse reward and long-horizon environments. The authors propose Temporal Distance-Aware Transition Augmentation (TempDATA), a novel method that generates additional transitions in a geometrically structured representation space rather than the state space. By learning state abstraction that captures temporal distance at both trajectory and transition levels, TempDATA enhances the ability to comprehend long-horizon behaviors efficiently. Experimental results demonstrate that TempDATA outperforms previous offline MBRL methods and achieves comparable or superior performance to diffusion-based trajectory augmentation and goal-conditioned RL across multiple benchmark environments, including D4RL AntMaze, FrankaKitchen, CALVIN, and pixel-based FrankaKitchen.

## update after rebuttal
The additional experiments on locomotion tasks improve the quality of this work. Currently, I have no additional questions. I would like to maintain my original rating, considering the overall algorithm novelty and contribution.

**Claims And Evidence:**

The algorithm design is validated by both theoretical and empirical analysis.

**Essential References Not Discussed:**

No missing essential references.

**Experimental Designs Or Analyses:**

The experiments in four different goal-achieving tasks

**Methods And Evaluation Criteria:**

The methods and evaluation metric are motivating.

**Other Comments Or Suggestions:**

See the Questions section.

**Other Strengths And Weaknesses:**

The theoretical proofs enhance the soundness of the proposed method, and the method design is motivating itself. However, the experiments are limited to only 4 goal-achieving tasks.

**Questions For Authors:**

1. The current experiments are conducted in four goal-achieving tasks. Is the proposed method applicable to non-goal-achieving tasks, such as locomotion tasks?

**Relation To Broader Scientific Literature:**

The paper is related to model-based trajectory augmentation in offline rl.

**Theoretical Claims:**

I checked the proofs presented in the appendix.

---

> ### Author Rebuttal · Authors · 2025-03-31
>
> We are grateful for the reviewer's thorough review and valuable suggestions about this work. Below, we outline how we have revised the paper to address the reviewer's concerns and questions.
>
> -----
> # Generalizability
> We acknowledge the importance of demonstrating applicability beyond long-horizon tasks. Following the reviewer's helpful suggestion, we ran additional experiments on dense-reward tasks from the widely recognized D4RL benchmark (e.g., halfcheetah and walker2d), achieving competitive performance as detailed below:
>
> | |MOPO (MBRL) |GTA (TARL) |IQL (MFRL) |TempDATA|
> |-|-|-|-|-|
> |halfcheetah-medium-replay|$68.2\pm20.8$|$50.0\pm0.8$|$43.4\pm0.5$|$62.8\pm8.9$|
> |halfcheetah-medium-expert|$63.3\pm38.0$|$93.1\pm3.1$ |$94.6\pm1.9$|$95.8\pm2.5$|
> |walker2d-medium-replay|$69.4\pm18.8$|$93.8\pm1.7$|$69.6\pm10.8$|$91.8\pm1.9$|
> |walker2d-medium-expert|$44.6\pm12.9$|$110.9\pm0.3$|$105.2\pm4.9$|$111.8\pm1.1$|
>
> These experiments demonstrate that the **proposed solution has competitive performance in widely used D4RL tasks** [1]. It further confirms our solution's effectiveness across various task types and reward structures.
>
> Additionally, given that our primary focus is on sparse-rewarded long-horizon tasks, we have presented strong empirical results across a diverse set of environments, e.g., AntMaze, FrankaKitchen, and CALVIN, highlighting our method's capabilities in subgoal navigation, manipulation, and decomposition tasks.
>
> -----
> We hope this response addresses your suggestion. Please let us know if any additional clarifications are required.
>
> # References
> [1] J. Fu, et al. D4RL: Datasets for deep data-driven reinforcement learning. arXiv preprint arXiv:2004.07219 (2020).

---

### Official Review · Reviewer_p9eS · 2025-03-23

**Overall Recommendation:** 2

**Summary:**

This paper presents an offline model-based reinforcement learning algorithm called TempDATA. The algorithm aims to tackle goal-conditioned tasks with long horizon and sparse task-completion reward.

----
The main idea is to learn an embedding space which that enables the computation of a temporal distance measure between pairs of states. The temporal distance allows the authors to perform reward shaping to decrease the distance to the goal.

Aside from reconstruction, the latent space is trained to also (1) satisfy temporal difference in the same trajectory, and (2) maintain proximity of consecutive states (Section 4.1). Next, the dynamics model learns to perform forward prediction in the latent space, and the synthetic rollouts are combined with the offline dataset for policy training.

----

The experiments consist of comparisons with various offline RL methods on a few domains, including state-based AntMaze, FrankaKitchen, Calvin and pixel-based Kitchen. The quantitative results demonstrate that (1) TempDATA outperform existing MBRL and TARL methods in the single-goal AntMaze environment. (2) TempDATA outperforms existing goal-conditioned offline RL methods in the multi-task kitchen and calvin environments. (3) TempDATA achieves similar performance as the state-of-the-art model-free offline RL method IQL.

----

## Update after Rebuttal

I appreciate the authors for sharing additional experiment results. The ablation study and new results in respond to other reviewers are sound.

I raised my score from 1 to 2 in recognition of the empirical results and the authors' explanations on the math formulations under my and reviewer DPTt's reviews. I did not raise my score further because I'm concerned that the paper might need substantial edits to clarify the math formulation.

**Claims And Evidence:**

"The latent dynamics model alleviates the overgeneralization issue with MBRL, and is more efficient than high-dimensional state space": this is only partially supported by the experiments. Yes, the offline RL results with the AntMaze is better than relevant baselines, but it is unclear whether the performance gain comes from synthetic transitions or the reward engineering trick. Moreover, the authors use a skill-conditioned policy. It is unclear how much this helped with training.

**Essential References Not Discussed:**

[Embed to Control: A Locally Linear Latent Dynamics Model for Control from Raw Images (NeurIPS 2015)] and [Accelerating Visual Sparse-Reward Learning with Latent Nearest-Demonstration-Guided Explorations (CoRL 2024)]: prior works have explored learning an embedding space and dynamics model to obtain a distance measure for reward shaping, although not in the offline MBRL setting.

**Experimental Designs Or Analyses:**

The choice of baseline methods for experimenting with single-goal, multi-goal and image-based settings are sound. However, the strongest offline MBRL results are only shown in the AntMaze environment. The other two settings don't show a clear advantage over the existing SOTA.

A key issue is that the algorithm has a lot of moving parts, including representation learning, dynamics learning, reward engineering and skill learning. Ablation studies are definitely needed to justify these design choices and understand their individual contributions.

**Methods And Evaluation Criteria:**

The evaluation domains are diverse. However, only one domain is used to test the main offline MBRL contribution.

**Other Comments Or Suggestions:**

The authors should consider making their core contributions more focused. TempDATA involves a lot of moving parts. For example, if the reward shaping  aspect is the most important/beneficial, they should compare with other value-based reward shaping methods.

**Other Strengths And Weaknesses:**

- The visualization in Figure 3 is helpful to justify the representation learning.

**Questions For Authors:**

- What is the unit for wall clock time in Figure 6(b)?
- Is $d$ just the L2 distance between $z$ vectors?
- How is GC-IQL performance better in pixel-based compared to state-based?

**Relation To Broader Scientific Literature:**

Offline RL and MBRL are hot fields. A new SOTA offline MBRL approach could be impactful. Additionally, a good pixel-based offline MBRL method could be a good baseline.

**Theoretical Claims:**

I checked the arguments in section 4.1 as well as in the appendix. The mathematical derivations are sloppy:
- In equation (1), $V = -d$ which makes sense because the value is higher the closer to the goal. However, in Theorem 4.2, the minus sign is dropped.
- In equation (2), the second row should also have a max over $\pi$.
- In $\mathcal{L}_{traj}$, the loss has $\mathcal{B}d + d$ but should be $\mathcal{B}d - d$.

---

> ### Author Rebuttal · Authors · 2025-03-31
>
> We appreciate the reviewer's valuable feedback.
>
> ---
> # Main contribution and reward shaping
> We agree with the reviewer’s observation that TempDATA involves multiple components and that it is important to delineate our main contribution clearly. Our main contribution lies in temporal-distance-aware autoencoder. This encoder serves as a unified representation module within our framework, enabling effective policy learning and data augmentation through temporal-distance-aware representations.
>
> Regarding the reward-shaping component, we clarify that our strategy follows widely used practices in skill-conditioned RL settings [1, 2, 3]. Distance-based reward shaping from encoded representations of state transitions is general. Therefore, our main contribution is not the reward shaping method itself but the accurate encoding of temporal distances between states, which empirically demonstrates a substantial improvement.
> ||Laplacian|Contrastive learning|Random|TempDATA|
> |-|-|-|-|-|
> |antmaze-large-play|$0\pm0$|$0\pm0$|$0\pm0$| $56.5\pm14.1$|
> |antmaze-large-medium|$0\pm0$| $0\pm0$|$0\pm0$|$44.2\pm15.3$|
> |kitchen-partial|$42.9\pm10.3$|$55.5\pm10.6$|$44.0\pm6.4$|$70.0\pm12.8$|
> |kitchen-mixed|$51.1\pm12.0$|$56.2\pm9.0$|$47.5\pm8.5$|$65.3\pm11.7$|
>
> To further substantiate the effectiveness of our solution, we ran additional experiments comparing TempDATA’s encoder with other representation-learning approaches. Specifically, we learn successor features with three alternative feature learners [4, 5] (Laplacian, Contrastive Learning, and Random Feature) as baselines, and compared their performance against TempDATA. The results demonstrate that our solution significantly outperforms alternative feature learners.
>
> ---
> # Generalizability
> Although our primary focus has been long-horizon tasks, we acknowledge the reviewer’s valid concern regarding generalization to broader task settings. To address the reviewer's concern, we ran additional experiments in dense rewarded task of D4RL. Our additional experiments demonstrate that the **proposed solution achieves performance comparable to widely used D4RL benchmark** (full details provided in response to Reviewer Akpp). Furthermore, we ran additional image-based experiments (see our detailed response to Reviewer DPTt), confirming that TempDATA performs well in visual domain, further supporting its generalizability.
>
> ---
> # Ablation study
> To alleviate the reviewer's concern, we conducted additional ablation experiments on AntMaze datasets with the following variants:
> - Skill-conditioned RL: without representation learning and dynamics learning
> - Skill-conditioned RL with MOPO: without representation learning
> - TempDATA with IQL: without reward engineering and skill learning
> | |Skill RL|Skill RL w MOPO|TempDATA w IQL|Full TempDATA|
> |-|-|-|-|-|
> |medium-paly|$66.5\pm14.6$|$21.9\pm13.2$|$76.0\pm10.4$|$74.8\pm8.3$|
> |medium-diverse|$65.9\pm11.2$|$26.0\pm10.8$|$65.5\pm8.9$|$69.5\pm10.8$|
> |large-play|$55.5\pm14.3$|$6.1\pm3.9$|$44.4\pm12.1$|$56.5\pm14.1$|
> |large-diverse|$49.0\pm18.8$|$9.3\pm6.0$|$47.5\pm9.5$|$44.2\pm15.3$|
>
> These results highlight the substantial contribution of the representation learning component, as performance noticeably decreases when it is removed (Skill RL with MOPO). Furthermore, even without reward engineering (TempDATA with IQL), the method maintains performance comparable to the full TempDATA, suggesting that our representation learning module plays a central role in TempDATA’s effectiveness.
>
> ---
> # Other questions
> 1. **Unit for wall clock time**.  We set the unit for wall clock time as hours in Figure 6 (b)
> 2. **How to define $d$ function**. We used Euclidean norm to calculate the distance between $z$.
> 3. **Pixel-based outperforms state-based**. In our experiments, GC-IQL with pixel-based tasks occasionally outperforms state-based ones. We believe this may be attributed to the strong performance of the IMPALA encoder [6], which can extract richer visual features that are not captured by low-dimensional state representations. We have also observed a similar experience in [7]; GCIQL, HIQL, and FQL are sometimes better in pixel-based than state-based tasks.
>
> We will update all references. We would be happy to continue the discussion if you have any other questions or comments that could raise your score.
>
> ---
> # References
> [1] S. Park, et al. Lipschitz-constrained unsupervised skill discovery. ICLR 2022.
>
> [2] K. Frans, et al. Unsupervised zero-shot reinforcement learning via functional reward encodings. ICML 2024.
>
> [3] R. Yang, et al. Behavior contrastive learning for unsupervised skill discovery. ICML 2023.
>
> [4] C. Zheng, et al. Contrastive difference predictive coding. ICLR 2024.
>
> [5] A. Touati et al. Does zero-shot reinforcement learning exist?. ICML 2023.
>
> [6] L. Espeholt, et al. IMPALA: Scalable distributed deep-RL with importance weighted actor-learner architectures. ICML 2018.
>
> [7] S. Park, et al. OGbench: Benchmarking offline goal-conditioned RL. ICLR 2025.

---

> > ### Comment · Reviewer_p9eS · 2025-04-05
> >
> > Thanks for answering my questions.
> >
> > Could the authors and/or other reviewers double check the math in section 4.1 and correct me if my initial concerns are wrong?

---

> > > ### Author Response · Authors · 2025-04-05
> > >
> > > We thank the reviewer for the careful examination of our theoretical completeness.
> > >
> > > ---
> > > We apologize for the typos and errors identified in the manuscript. Due to the 5,000-character limit of the initial rebuttal, we were unable to include detailed responses to this specific issue earlier. We have considered your suggestions and decided to reflect them clearly in our revised manuscript as follows:
> > >
> > > **Regarding equation (1):**
> > >
> > > We recognize the potential for confusion regarding the sign used in our expression. We will explicitly clarify and add a sign to avoid possible misunderstandings.
> > >
> > > **Regarding equation (2):**
> > >
> > > While we intended $max_{a \in \mathcal{A}} to implicitly represent the optimal Q-function (thus implicitly maximizing over policies), we acknowledge that our original expression could indeed lead to confusion. Although we stated above this equation, "Therefore, the optimal goal-conditioned value function equals the Q-function with an optimal policy," we recognize the need to modify this equivalence directly in the equation by following the reviewer's suggestion. We will revise the manuscript accordingly to indicate explicitly that the optimal policy is considered.
> > >
> > > **Regarding $\mathcal{L}_{traj}$:**
> > >
> > > As correctly identified, our Bellman operator $\mathcal{B}d$ was explicitly defined as $\mathcal{B}d = -r - Q'$ style. Thus, the trajectory loss $\mathcal{L}_{traj}$ using the form $\mathcal{B}d + d$ is internally consistent and correct within our defined theoretical framework. However, recognizing conventional practices and potential sources of confusion for readers, we agree with your recommendation to revise the notation to follow the traditional Bellman operator convention clearly. We will adjust the signs accordingly to reflect standard conventions explicitly.
> > >
> > > ---
> > > Once again, we appreciate your careful review, which has helped us improve the clarity of our paper. If there are any further questions or additional points you would like to discuss, we would be more than happy to continue the discussion!
> > >
> > > ---
> > > **Update Apr 07** We appreciate that the reviewer has adjusted the score from 1 to 2, recognizing our efforts to address your concerns. However, we would gently like to inquire about any remaining concerns or reasons that might still lean your decision toward rejection. Your clarification on this matter would greatly assist us in potentially improving this work for addressing reviewers' concerns and further submission!

---

### Decision · Program_Chairs · 2025-05-01

**Decision:**

Accept (poster)

**Comment:**

This paper was discussed controversially and even after rebuttal and discussion the assessments ranged from Weak Reject to Accept. The rebuttal was rated as very strong and the results of the two additional experiments are really impressive.

The criticism remains that the initial submission did not reach this level of quality and that extensive changes were necessary and were made, but can no longer be assessed by the reviewers, and that substantial edits are needed to clarify the math formulation.